# How do healthcare professionals on non-palliative care wards perceive quality of care in the dying phase? Personal and organizational predictors identified in a cross-sectional study

Nikolas Oubaid[1]*, Sukhvir Kaur[2], Karin Oechsle[1], Viola Milke[1], Anneke Ullrich[1], Aneta Schieferdecker[1], Kerstin Kremeike[2], Sophie Meesters[2], Christin Herrmann[3], Raymond Voltz[2,4,5], Holger Schulz[6]

1 Palliative Care Unit, Department of Oncology, Hematology and BMT, University Medical Center Hamburg-Eppendorf, Hamburg, Germany, 2 Department of Palliative Medicine, Faculty of Medicine and Cologne University Hospital, University of Cologne, Cologne, Germany, 3 Chair of Quality Development and Evaluation in Rehabilitation, Institute of Medical Sociology, Health Services Research and Rehabilitation Science, Faculty of Human Sciences & Faculty of Medicine and Cologne University Hospital, University of Cologne, Cologne, Germany, 4 Center for Integrated Oncology Aachen Bonn Cologne Duesseldorf (CIO ABCD), Faculty of Medicine and Cologne University Hospital, University of Cologne, Cologne, Germany, 5 Center for Health Services Research (ZVFK), Faculty of Medicine and Cologne University Hospital, University of Cologne, Cologne, Germany, 6 Department of Medical Psychology, University Medical Center Hamburg-Eppendorf, Hamburg, Germany

* ni.oubaid@uke.de

## Abstract

### Background

Most people in European countries die in hospitals outside of specialist palliative care wards. Healthcare professionals of all disciplines are therefore often involved in the care for dying patients. Healthcare professionals' perception of quality of care in the dying phase as well as its predictors are of interest to improve quality of care on non-palliative care hospital wards.

### Aim

Identification of personal and organizational predictors of healthcare professionals' perceived quality of care in the dying phase.

### Methods

Cross-sectional online survey with healthcare professionals of ten non-palliative care hospital wards of two university medical centers. Descriptive statistics were used to describe the data. A hierarchical linear regression model with ten theoretically derived personal (gender, age, profession, palliative care training, spirituality, two self-care items, general self-efficacy, thanatophobia, burden factors when caring for dying patients) and two organizational predictors (type of ward, interprofessional

**Data availability statement:** Data cannot be shared publicly because participants were guaranteed protection of personal data within the confines of the European data protection act (General Data Protection Regulation (GDPR)) and the data contains potentially identifying information. In consultation with the staff councils of the participating university medical centers, the informed written consent additionally assured only aggregated data would be published. However, Data are available from: University of Cologne, Medical Faculty, Department of Palliative Medicine, Kerpener Straße 62 50937 Cologne, Germany e-mail: innofonds-sterbephase@uk-koeln.de for researchers who meet the criteria for access to confidential data.

**Funding:** The project was funded by the German Innovation fund, Federal Joint Committee (G-BA, 01VSF19033). The funding body plays no role in the design of the study and collection, analysis, and interpretation of data and in writing the manuscript. We acknowledge financial support from the Open Access Publication Fund of UKE - Universitätsklinikum Hamburg-Eppendorf.

**Competing interests:** The authors have declared that no competing interests exist.

**Abbreviations:** HCP, Healthcare professional; ICU, Intensive care unit; VIF, Variance inflation factor.

patient-centered teamwork) was developed. The dependent variable was an eleven-point Likert-scaled item (0 = extremely bad, 10 = ideal) measuring the quality of care in the dying phase at the respective ward, perceived by healthcare professionals. Predictors were categorized as modifiable and non-modifiable.

## Results

Most of the n = 201 participants were female (64.7%), nurses (57.2%) and 30–50 years old (53.2%). The regression model was statistically significant (p < 0.001) and explained 30.7% of the total variance. Lower perceived quality of care in the dying phase was associated with younger age (β = 0.15, ρ = 0.020), being a nurse (β = 0.29, ρ < 0.001), and lower perception of interprofessional patient-centered teamwork on their ward (β = 0.37, ρ < 0.001).

## Discussion

Perceived quality of interprofessional patient-centered teamwork was the most clinically relevant predictor in this model, as it had the strongest association and was modifiable. Age and profession were significant, non-modifiable predictors but can be considered when implementing interventions. As improving the perceived quality of care in the dying phase could be beneficial for dying patients, interventions strengthening interprofessional patient-centered teamwork should be implemented on non-palliative care hospital wards.

## Introduction

Research showed that the quality of care for dying patients in hospitals has potential for improvement [1,2,3]. To ensure the most efficient use of resources to improve the quality of care for dying patients, it is necessary to know which specific factors have a relevant association with the quality of care in the patients' dying phase before selecting and implementing interventions. Previous studies have already assessed some healthcare professionals (HCPs)-related factors with potentially predictive character for general quality of care and the broader end-of-life care:

Regarding HCPs' gender, studies on the general quality of care and end-of-life care showed no significant difference between the quality of care provided by female and male HCPs [4,5]. A systematic review showed, that there is no relevant difference between physicians and nurses in the quality of care they provide [6]. Personal self-efficacy (p < 0.001) and professional self-efficacy (p < 0.01) are reported to have positive (bivariate) correlations with HCPs' job performance [7,8]. Occupational stress and little self-care are dysfunctional and can lead to burnout in HCPs [9–11], which is associated with a lower quality of care and patient safety and a decreased ability to work [9,12,13]. Studies and reviews of the literature have consistently shown that interprofessional teamwork – consisting of team interaction and mutual communication, collaborative practice and shared goals, and interprofessional competencies in

HCPs - is an essential part of care and has a positive impact on the quality of care perceived by HCPs, patients and their informal caregivers in palliative care and general care settings [14–18].

In addition to the personal predictors mentioned above, the setting of care is also relevant. Hospital care was described as unsatisfactory by informal caregivers [2]: Intensive care units (ICUs) in particular were rated worse than hospice settings or palliative care units with respect to sensitive communication, relief of pain and other symptoms and collaboration with other services [2,19]. ICUs have also been particularly challenged in providing care in the dying phase [20,21] (e.g., regarding decision-making and determining prognosis [22]) and informal caregivers of patients who died in ICUs also carried a higher risk of post-traumatic stress compared to informal caregivers of patients who died at home or in hospice setting [19].

In addition, there are other factors that are being discussed regarding whether and to what extent they are predictive of quality of (end-of-life) care (e.g. HCPs' own religious values [5,23], HCPs' age and clinical experience [24,25], HCPs' fear of death and dying (thanatophobia) [26], palliative care-specific training for HCPs [27]).

Many studies focused on quality of care reported by patients and informal caregivers. However, HCPs' perception represents also an important indicator for quality of care in the patients' dying phase as their perception is less influenced by personal emotions: Dying patients are often non-responsive [28], have or develop cognitive impairment [29] during the course of their disease or face other communication challenges in HCP-patient interaction [30,31]. As a result, an external assessment of the quality of care is needed, which can be provided by HCPs [30,32]. The perception of HCPs is therefore an important parameter to ensure good quality of care (e.g., pain assessment and management) for dying patients [33,34].

Our research interest was to assess whether the discussed predictors of quality of care and end-of-life care are associated with HCPs' perceived quality of care for dying patients in an exploratory model Our study focused specifically on the patients' dying phase (last 3–7 days [35]) and not the broader end-of-life care. As most hospital patients in Germany die outside of palliative care wards [36], we focused on ICUs and general wards only. We surveyed a multiprofessional cohort of HCPs and included general wards – a clinical setting that is little researched regarding quality of care in the dying phase. Our specific exploratory research question was: What personal and organizational factors predict HCPs' perceived quality of care in the dying phase on non-palliative care hospital wards?

## Methods

### Study design

The survey was designed as a cross-sectional online survey within the context of a multi-center bottom-up intervention project that pursues the idea of how the quality of care for dying patients on non-palliative care wards can be improved by developing ward-specific measures at the University Medical Centers Cologne and Hamburg-Eppendorf, Germany [3].

The underlying survey collected the assessment of HCPs of wards participating in our project on the following topics: (Table 1).

### Description of the scales

The German version of the General Self-Efficacy Scale by Schwarzer and Jerusalem [39] rates ten items on a four-point Likert scale with (1) "not at all true" to (4) "exactly true". Reliability is α = 0.80–0.90 for German-speaking samples. The results can be summarized into a total score. High values indicate high self-efficacy.

A German version [42] of the thanatophobia scale by Merrill et al. [41] measures fear of death and dying in HCPs when caring for dying patients on a seven-point Likert scale. Each of the seven items is rated from (1) "strongly disagree" to (7) "strongly agree". Reliability is α = 0.82–0.87 in the original study. The results can be transformed into a total score. High values indicate high thanatophobia.

**Table 1. Overview of the questionnaire.**

| Section | Topic/questionnaire | Number of items (response scale) |
|---|---|---|
| 1 | Sociodemographic data including spiritual aspects (self-constructed) | 7 (mixed) |
| 2 | Professional background and experience regarding death and dying on the specific ward (self-constructed) | 3 (mixed) |
| 3 | Perceived quality of care in the dying phase (self-constructed) | 1 (11-point Likert) |
| 4 | Burden factors related to care in the dying phase [37] | 11 (4-point Likert) |
| 5 | General Self-Efficacy Scale [38,39] | 10 (4-point Likert) |
| 6 | Self-assessment in dealing with dying patients and their ICs [40] | 19 (5-point Likert) |
| 7 | Thanatophobia Scale [41,42] | 7 (7-point Likert) |
| 8 | Self-care [43] | 3 (mixed) |
| 9 | Internal Participation Scale [44] | 6 (4-point Likert) |

Abbr.: IC = informal caregiver.

The Internal Participation Scale by Körner and Wirtz [44] measures interprofessional patient-centered teamwork (consisting of working climate, cooperation, agreements, coordination, communication and respect) on a four-point Likert scale. Six items are rated from (1) "does not apply at all" to (4) "fully applies", with the further possibility of selecting an "I can't judge this" category. Reliability is α = 0.87 in the original study. The results can be summarized into a total score ranging from 0–100. High values indicate high quality of interprofessional patient-centered teamwork.

Burden related to care in the dying phase is measured by items adapted from Müller et al. [37]. The eleven items use a four-point Likert scale to measure the intensity of burden factors in the care for dying patients and their informal caregivers in HCPs. The response options range from (1) "not burdened at all" to (4) "very severely burdened". As the authors do not give recommendations for a specific method to summarize the items, we created a sum score that can theoretically take on values from 11 to 44. High values indicate a high burden related to care in the dying phase. A detailed assessment of the burden factors can be found elsewhere [45].

## Data collection

This online survey was pilot tested by members of the research team and a convenience sample of n = 6 multiprofessional HCPs. Ethical approval was obtained from the ethics committee of the Medical Faculty of the University of Cologne on the 19th of April 2021 (20-1727) and by the ethics committee of the General Medical Chamber Hamburg on the 3rd of August 2021 (2021-200061-BO-bet). The approval of all relevant staff councils of the two university medical centers was received. The survey had an information page about the aim of the study and data privacy. By clicking on the "continue" button, participants gave their written informed consent to take part in the survey. We did not perform a power analysis for this specific study but expected a response rate of ≥50% [3].

The anonymous online survey was administered in ten hospital wards (four general wards, six ICUs) of the two university medical centers participating in our project [3]. HCPs of these wards were recruited by the ward leadership. Inclusion criteria were the assessment of the ward heads as to whether the HCP was part of the team and sufficient knowledge of the German language. The survey link was sent by e-mail from the research team to the ward leaderships (physicians and nurses) who then sent the survey to the respective team members via internal mailing lists within two to three days. Additionally, posters with QR-codes directing to the survey were hung on the respective wards. Data was collected between 6th of September and 10th of December 2021, including two reminders to increase participation. Data collection was during the COVID-19 pandemic. The median time to complete a questionnaire was 9 minutes and 36 seconds. The response rate was 35% (a post-hoc power analysis can be found in the S2 Appendix). Missing values and time stamps were checked during data processing to exclude non-valid results.

*LimeSurvey* was used as an online tool for data collection. Further information regarding study design, survey construction and data collection can be looked up in the S1 and S2 Appendices (CHERRIES-Guideline [46] and STROBE-Guideline [47]).

## Data analysis

Data were explored and examined using descriptive statistics and dichotomized where necessary. A hierarchical regression model was set up including ten personal variables (gender, age, occupation, spirituality, any palliative training, self-care (2x), general self-efficacy, burden related to care in the dying phase and thanatophobia), and the two organizational variables (perceived quality of interprofessional patient-centered teamwork and type of ward) were introduced in a second step. We intentionally refrained from using available standardized complex item sets to measure the quality of care in the dying phase [30] in our non-palliative care setting to avoid the impression of quality control, keep the questionnaire short and increase the completion rate of the survey. Deviations of the dependent variable from a normal distribution were visually evaluated with a Q-Q plot and homoscedasticity and distribution of the residuals were visually checked by residual plots [48]. For a well interpretable model, the following statistical parameters were used as cutoff values: The variance inflation factor (VIF) – which measures multicollinearity between the predictors – is recommended to be less than five [49]. We used Cook's distance to check for outliers – cases with values above 1.00 were rated as outliers [50]. Alpha-levels <0.05 were considered statistically significant. We calculated Akaike Information Criterion to compare both models [51]. We used adjusted $R^2$ and change in $R^2$ to determine model strength and for our research question and the number and type of predictor variables we expected a good model – according to Cohen [52] – to have an adjusted $R^2$ value of at least 13%. Survey with missing values were excluded. *IBM SPSS V27* was used for data processing and analysis.

For better interpretation of the results, we categorized predictors as "modifiable" or "non-modifiable". We categorized a predictor as "modifiable" if we believe that ward leadership can (positively) alter the predictor.

## Results

### Participants

Among the n = 201 participants, most were female HCPs (64.7%). More than half of the HCPs were aged between 30 and 50 years (53.2%) and worked in their profession for 6–20 years (43.3%). Most HCPs had vocational training in their profession (60.8%) and most HCPs were nurses (57.2%). Most were intensive care staff (62.7%). Table 2 shows the specific characteristics of the participants.

For further exploratory analysis in a regression model, two variables were dichotomized: Profession was dichotomized into "nurse" (n = 115, 57.2%) and "other than nurse" (n = 86, 42.8%) and age was dichotomized into "≤50 years" (n = 167, 83.1%) and ">50 years" (n = 34; 16.9%). We used "nurse" and ">50 years" as reference categories to compare younger and older HCPs and have nurses stand alone to have a balanced sample.

### Descriptive results

The distribution and descriptive statistics for HCPs' perceived quality of care in the dying phase are shown in Fig 1. Quality of care in the dying phase was perceived by HCPs with M = 5.0 [4.7, 5.3] and SD = 2.2 (scale: 0–10). According to Fig 1, a left-skewed distribution of the data has been found. After inspection with the Q-Q plot, the distribution was considered acceptable for the regression analysis. Further statistical details are listed in the S3 Appendix (Analysis Appendix).

We used different scales to assess attitudes regarding self-efficacy, thanatophobia, interprofessional patient-centered teamwork and burden factors when caring for dying patients. In our sample, the Cronbach's α of the standardized scales were acceptably high and comparable to the primary studies, and the reliability of the non-validated burden scale was also acceptable with α = 0.79 (see Table 3). A visual presentation of the data (frequencies, boxplots and Q-Q-diagrams) can be found in the S3 Appendix (Analysis Appendix).

To address spirituality and self-care, three self-constructed items were used (see Table 4).

**Table 2. Sociodemographic data of the participants: n = 201.**

| Participants characteristics | GWs (n = 75) | | ICUs (n = 126) | | Total (n = 201) | |
|---|---|---|---|---|---|---|
| | n | % | n | % | n | % |
| Gender | | | | | | |
| Female | 54 | 72.0 | 76 | 60.3 | 130 | 64.7 |
| Male | 21 | 28.0 | 50 | 39.7 | 71 | 35.3 |
| Age (years) | | | | | | |
| < 30 | 24 | 32.0 | 36 | 28.6 | 60 | 29.9 |
| 30-50 | 38 | 50.7 | 69 | 54.8 | 107 | 53.2 |
| > 50 | 13 | 17.3 | 21 | 16.7 | 34 | 16.9 |
| Profession | | | | | | |
| Physician | 20 | 26.7 | 34 | 27.0 | 54 | 26.9 |
| Nurse | 34 | 45.3 | 81 | 64.3 | 115 | 57.2 |
| Psychosocial care/Chaplain/Therapist [a] | 16 | 21.3 | 9 | 7.1 | 25 | 12.4 |
| Other | 5 | 6.7 | 2 | 1.6 | 7 | 3.5 |
| Professional experience (years) | | | | | | |
| < 1 | 4 | 5.3 | 6 | 4.8 | 10 | 5.0 |
| 1-5 | 21 | 28.0 | 40 | 31.7 | 61 | 30.3 |
| 6-20 | 31 | 41.3 | 56 | 44.4 | 87 | 43.3 |
| > 20 | 19 | 25.3 | 24 | 19.0 | 43 | 21.4 |
| Education[b] | | | | | | |
| No completed education in profession[c] | 0 | 0.0 | 2 | 1.6 | 2 | 1.0 |
| Vocational education in profession | 45 | 60.0 | 82 | 65.1 | 127 | 60.8 |
| University education in profession | 34 | 45.3 | 46 | 36.5 | 80 | 38.3 |
| Palliative care training (any)[d] | | | | | | |
| Yes | 26 | 34.7 | 52 | 41.3 | 78 | 38.8 |
| No | 49 | 65.3 | 74 | 58.7 | 123 | 61.2 |

[a]e.g., physical therapist,

[b]multiple responses possible,

[c]e.g., nurses and physicians in training,

[d]any training in palliative medicine/palliative care, regardless of the scope and degree/certificate.

Abbr.: GW = general ward; ICU = intensive care unit.

## Hierarchical multiple regression

Step 1 included ten personal variables, and the explained variance was 21.4% (adj. $R^2 = 0.171$, $p < 0.001$). Further information regarding the step 1 model can be found in the S3 Appendix (Analysis Appendix). The overall model included two additional organizational variables, and the explained variance was 35.0% (adj. $R^2 = 0.307$, $p < 0.001$). The change in $R^2$ was $\Delta = 0.136$. Akaike Information Criterion values were 286.8 (step 1) and 254.1 (overall model). Multicollinearity of the overall model was checked by VIF and took values between 1.13 and 1.87, indicating the absence of multicollinearity. Homoscedasticity and normal distribution of the residuals were checked visually and found to be present. No outliers were identified in the analysis (Cook's distance values ranged between <0.01 and 0.11). The statistical requirements for a multiple regression model were therefore considered acceptable. Detailed results are shown in Table 5.

In the overall model, three variables showed a significant association, controlled for the presence of all other predictors, with the perceived quality of care in the dying phase: HCPs who were older than 50 years perceived the quality of care in the dying phase on their ward better than younger ones. Nurses perceived the quality of care in the dying phase on their

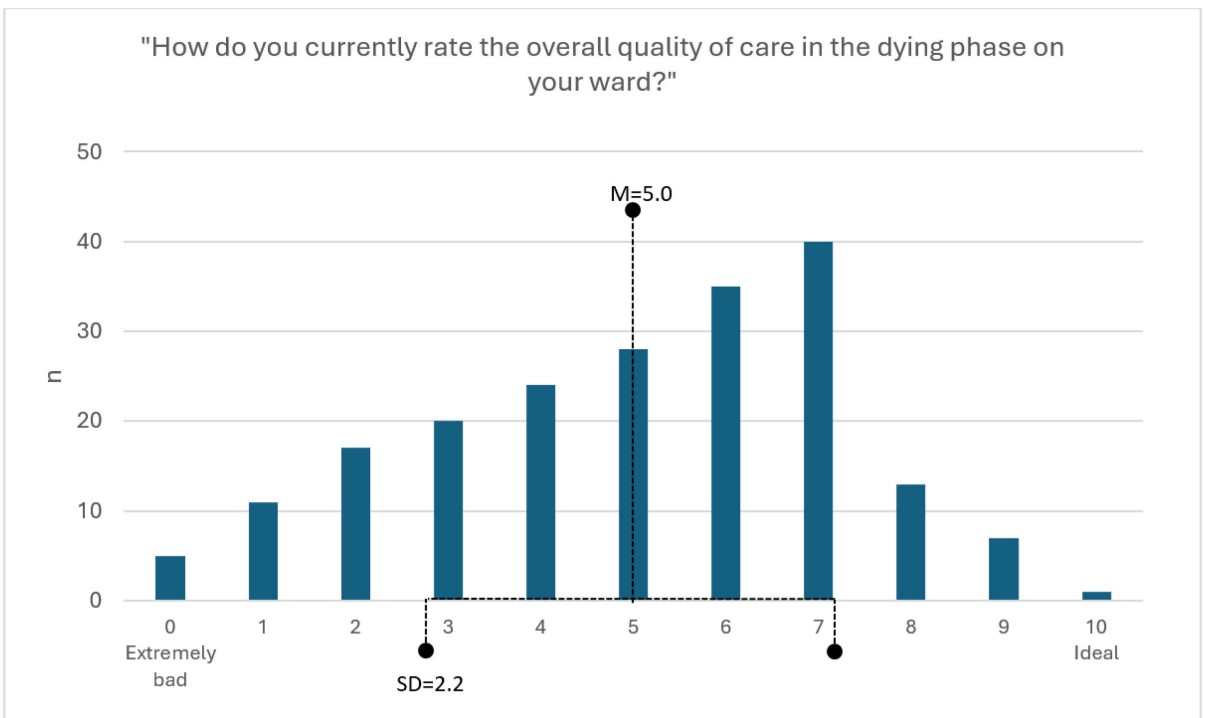

**Fig 1. Frequency distribution of HCPs' perceived quality of care (n=201).** Abbr.: M=mean, SD=standard deviation, HCP=healthcare professional.

**Table 3. Descriptive statistics of the standardized questionnaires.**

| Scale | Theoretical range (empirical range) | Cronbach's α | M [95% CI LL, UL] | SD |
|---|---|---|---|---|
| General self-efficacy (n=201) | 10-40 (17-39) | 0.82 | 28.5 [27.9, 29.0] | 3.8 |
| Thanatophobia (n=201) | 7-49 (7-41) | 0.86 | 16.5 [15.4, 17.6] | 8.0 |
| Internal participation (n=193)[a] | 0-100 (0-100) | 0.88[b] | 71.3 [68.8, 73.8] | 17.4 |
| Burden factors (n=201) | 11-44 (11-36) | 0.79 | 21.8 [21.1, 22.5] | 4.7 |

[a]n is less than 201 due to the "I can't judge this" category, which was excluded.

[b]n=190.

Abbr.: M=mean, SD=standard deviation, CI=confidence interval LL=lower limit, UL=upper limit.

ward worse than other HCPs. The better HCPs perceived the quality of interprofessional patient-centered teamwork on their ward, the better they perceived the quality of care in the dying phase. The second organizational variable (type of ward) just failed to reach significance (p=0.060). There were no interaction effects between type of ward and profession (p=0.741, also see S3 Analysis Appendix).

Table 6 shows our applied grouping of the predictor variables to personal and organizational variables, as well as if these predictors are modifiable. Among the statistically significant predictors, interprofessional patient-centered teamwork is the only modifiable predictor that can be influenced by the ward leadership.

**Table 4. Descriptive statistics of the three items on self-care and religion (n = 201).**

| Content | Item | Range | M [95% CI LL, UL] | SD |
|---|---|---|---|---|
| Spirituality | How much do religiosity and/or spirituality influence your professional actions? | (1) Not at all – (5) very much | 2.3 [2.2, 2.5] | 1.0 |
| Self-care (delimit) | Regarding the treatment for dying patients, I know strategies to delimit myself. | (1) Does not apply at all – (5) fully applies | 3.6 [3.5, 3.7] | 0.9 |
| Self-care (support opportunities) | I have support options when I reach my limits. | (1) Does not apply at all – (5) fully applies | 3.8 [3.6, 3.9] | 1.1 |

Abbr.: M = mean, SD = standard deviation, CI = confidence interval, LL = lower limit, UL = upper limit.

**Table 5. Results of the hierarchical multiple regression (n = 193).**

| Overall model (adj. $R^2$ = 30.7%) | Modifiable? | B | 95% CI LL | 95% CI UL | β | ρ |
|---|---|---|---|---|---|---|
| Personal variables | | | | | | |
| Gender[a] | No | 0.12 | −0.50 | 0.74 | 0.03 | 0.702 |
| Age[b] | No | 0.94 | 0.15 | 1.73 | 0.15 | 0.020* |
| Profession[c] | No | −1.31 | −1.91 | −0.71 | −0.29 | <0.001* |
| Palliative care training (any)[d] | Yes | −0.49 | −1.10 | 0.12 | −0.11 | 0.115 |
| Spirituality | No | 0.12 | −0.18 | 0.41 | 0.05 | 0.437 |
| Self-care (delimit) | Yes | −0.07 | −0.48 | 0.34 | −0.03 | 0.751 |
| Self-care (support opportunities) | Yes | 0.22 | −0.10 | 0.54 | 0.10 | 0.182 |
| General self-efficacy | Yes | 0.01 | −0.07 | 0.09 | 0.02 | 0.821 |
| Thanatophobia | Yes | −0.02 | −0.06 | 0.03 | −0.05 | 0.464 |
| Burden factors | Yes | −0.01 | −0.08 | 0.06 | −0.02 | 0.841 |
| Organizational variables | | | | | | |
| Type of ward[e] | No | 0.58 | −0.03 | 1.18 | 0.12 | 0.060 |
| Interprofessional teamwork | Yes | 0.05 | 0.03 | 0.07 | 0.37 | <0.001* |

[a]Female = 1, male = 2,

[b]≤ 50 = 0, > 50 years = 1,

[c]Other than nurses = 0, nurse = 1,

[d]No = 0, yes = 1,

[e]Intensive care unit = 1, general ward = 2.

* = statistically significant for p < 0.05.

Abbr.: adj. = adjusted, CI = confidence interval, LL = lower limit, UL = upper limit.

## Discussion

The perception of quality of care in the dying phase from the perspective of HCPs is an important aspect in evaluating and improving the quality of care for dying patients and the working environment in clinical settings. This cross-sectional, exploratory study has shown which of the discussed predictors of quality of care reported by patients and informal caregivers are also predictors of HCPs' perceived quality of care in the dying phase. It was shown that older age, non-nursing profession and a more positive perception of interprofessional patient-centered teamwork on the ward are predictors of HCPs' perceived quality of care in the dying phase on non-palliative care hospital wards.

**Table 6. Clustering of the 12 predictor variables into personal and organizational, as well as modifiable and non-modifiable groups from leadership perspective.**

|  | *Modifiable* | *Non-modifiable* |
|---|---|---|
| *Personal determinants* | Self-care (delimit strategies, support) | Gender |
|  | Self-efficacy | Spirituality |
|  | Thanatophobia | Age*<br>(β = 0.15; weak association) |
|  | Burden related to care in the dying phase | Profession***<br>(β = −0.29; weak association) |
|  | Palliative training (any) |  |
| *Organizational determinants* | Interprofessional patient-centered teamwork***<br>(β = 0.37; moderate association) | Type of ward |

Statistically significant for *$p < 0.05$ and ***$p < 0.001$.

## Discussion of significant predictors

**Interprofessional patient-centered teamwork.** The strongest association with the perceived quality of care in the dying phase was interprofessional patient-centered teamwork in this analysis. The better HCPs perceived the quality of interprofessional patient-centered teamwork on their ward, the better they perceived the quality of care in the dying phase. This finding is in line with previously mentioned studies and reviews of the literature in the background chapter that have examined aspects of multiprofessional teamwork in relation to overall quality of care, care safety and end-of-life care. This moderate association with HCPs' perception of the quality of care in the dying phase proves the necessity of interprofessional collaboration, shared goals of care and interprofessional knowledge to improve the quality of care for patients in the dying phase.

**Age.** Older HCPs perceived the quality of care in the dying phase on their ward better than younger ones. Older HCPs are more experienced and generally less prone to stress and burnout [53,54], which is known to be associated with lower perceived overall quality of care and patient safety. Another possible explanation might be related to HCPs' work experience and the degree of their perceived "moral distress". Moral distress refers to the negative emotional reaction of HCPs when they feel that they "(...) have lost some of their personal and professional integrity, have been compromised as a moral agent in practicing in accordance with accepted values and standards, or have abandoned their ethical principles" (Kherbache et. al 2022: 1972) [55]. A possible explanation for the discrepancy in perception of quality of care between younger and older HCPs could be that younger HCPs compare the care provided on their ward with a more idealistic care they know from their training and education. Studies suggest that moral distress is significantly higher in younger HCPs [56,57] and that older HCPs experience moral distress less intensively [55]. However, methodologically it should be noted that this predictor had a small effect size and a relatively wide 95% CI in our data. This finding therefore fits into the discussion of whether age is a relevant predictor of quality of care.

**Profession.** Nurses perceived the quality of care in the dying phase worse than other professions. This result is consistent with comparable studies, which observed a similar effect for critical care [58,59]. There are several possible explanations for this result: One would be that nurses spend more time with patients and their informal caregivers than HCPs of most other professions [31,58,60] and that care for the dying is traditionally their task [61]. Therefore, nurses might have a closer relationship with the dying and might recognize the signs of a beginning dying phase earlier than other professions [31,62]. Another explanation might be related to communication barriers between different professions [18,63,64], consequently uncertainties about everyone's scope of duties [62] and a resulting information deficit and therefore different perception of quality of care. Finally, in Germany, the responsibility for medical decision-making lies only with physicians [64]. Nurses are therefore able to criticize the decisions of others more easily.

## Discussion of non-significant predictors

Gender, palliative training, spirituality, self-care, self-efficacy, thanatophobia, burden factors and the type of ward were not significantly associated with the perceived quality of care in the dying phase in our data. Some of these results are in line with the heterogeneous results from previous studies (e.g., thanatophobia or palliative training). The lack of a statistically significant association between the burden score and perceived quality of care in the dying phase could be due to the chosen measurement instrument and scoring method. The lack of a statistically significant association between ward type and perceived quality of care in the dying phase is particularly noteworthy here, as it falls just short of the prespecified level of significance. It can be assumed that this difference could be statistically significant in a larger and more balanced sample, as the topic of death and dying is of particular relevance in ICUs [26,31,65]. Some previous studies have exclusively used bivariate correlations regarding these predictors and may therefore have less relevance in our regression model. It can also be assumed that the discrepancy results from the fact that most previous studies measured certain aspects of quality of (end-of-life) care (e.g., psychological care, symptom relief) and did not assess the perception of HCPs but used other dependent variables (e.g., patient or informal caregiver assessment). Nevertheless, the self-constructed items must be checked for their validity. The response rate (35%) may have been biased because potential participants knew that this survey was part of a project to improve the quality of care in the dying phase on their ward. It is therefore possible that the most motivated HCPs responded to the survey. However, as the distribution of HCPs' perceived quality of care did not show any abnormalities and the regression did not show any outliers, a response bias cannot be assumed.

## Strengths and limitations

A strength of this study is the simultaneous test of several personal and organizational variables as predictors for HCPs' perceived quality of care in the dying phase. Additionally, this study includes the views of multiprofessional HCPs (e.g., nurses, physicians, psychosocial caregivers) of non-palliative care hospital wards, which is crucial for further improving the quality of care in the dying phase in the hospital setting. Especially the perception of HCPs of general wards is underrepresented in previous studies. However, there are also several limitations to this work: The preponderance of ICUs, which have a higher number of deaths compared to general wards, must be recognized as a potential selection bias as more than 60% of our sample were ICU staff. Also, there is an underrepresentation of non-nurses in this sample, especially in ICUs. There was a low response rate, which may have slightly biased the results. In this context it must also be noted that this study was done during the COVID-19 pandemic and it must be assumed that hospital wards, especially ICUs, were especially challenged during this time. In addition, other organizational factors, such as staffing and resources, management and leadership style, and personal factors, such as HCP job satisfaction, competence or conscientiousness, were not assessed. Finally, the validity of the self-constructed and translated items and the burden factors need to be further examined.

## Clinical implications

This study showed that the concept of interprofessional patient-centered teamwork had a significant and clinically relevant association with HCPs' perceived quality of care in the dying phase. Age and profession were also significantly associated with perceived quality of care in the dying phase but are not modifiable by ward leadership. However, they should still be taken into consideration by ward leadership when implementing interventions to improve care in the dying phase. Interventions introduced by ward leadership could be multiprofessional case or team meetings, multiprofessional discussions with patients and informal caregivers, or supervision with special attention to the profession and age of HCPs. Miller et al. list 14 different team-building interventions for non-acute hospital settings in their systematic review [66] (e.g., TeamSTEPPS [67]). Weller et al. provide suggestions for improving communication between different professions in healthcare teams (e.g., structured information transfer and protocols) [18]. Ward leadership can therefore draw on already

established training and communication programs to increase the quality of interprofessional patient-centered teamwork among HCPs on their wards. These team-based approaches, with special attention paid to HCPs of younger age and nurses, could improve the perceived quality of care in the dying phase and also have positive secondary outcomes (e.g., stress reduction among HCPs [54] and job satisfaction [68]).

### Future research

Because of the exploratory character of this study, further research is needed. The results of this study are partly in line with the results of comparable studies, although the referenced studies mostly refer to general quality of care, the more abstract term of end-of-life-care and mostly used patients' and informal caregivers' perceptions. Despite a high level of explained variance of our model, other important predictors remain unknown. The model needs to be tested in an independent, larger sample, outside of a pandemic context and with a more balanced distribution regarding type of ward and profession. Also, more organizational predictors (e.g., staffing, deaths per unit time or resources of the ward) should be included. It could also be of interest to evaluate if the perceived quality of care correlates with moral distress in HCPs and whether perceived quality of care increased after the intervention phase of the project [3].

## Supporting information

**S1 Appendix: CHERRIES-Guideline: Includes additional information regarding the survey construction and the data collection process.**
(PDF)

**S2 Appendix: STROBE-Guideline: Includes additional information regarding the cross-sectional design of the study.**
(PDF)

**S3 Appendix: Analysis Appendix: Includes additional information regarding statistical analysis (the minimal regression model, correlation matrix of model variables, test for interaction between type of ward and profession and visual data presentation of used index-variables).**
(PDF)

## Acknowledgments

We would like to thank the leadership of the participating hospital wards and their healthcare professionals for making this research possible by taking the time to complete our survey.

## Author contributions

**Conceptualization:** Nikolas Oubaid, Karin Oechsle, Anneke Ullrich, Holger Schulz.

**Data curation:** Nikolas Oubaid, Sukhvir Kaur.

**Formal analysis:** Nikolas Oubaid.

**Funding acquisition:** Karin Oechsle, Kerstin Kremeike, Raymond Voltz.

**Investigation:** Nikolas Oubaid, Holger Schulz.

**Methodology:** Nikolas Oubaid, Sukhvir Kaur, Anneke Ullrich, Holger Schulz.

**Project administration:** Karin Oechsle, Kerstin Kremeike, Raymond Voltz.

**Resources:** Karin Oechsle, Anneke Ullrich, Kerstin Kremeike, Raymond Voltz.

**Software:** Nikolas Oubaid.

**Supervision:** Karin Oechsle, Holger Schulz.

**Validation:** Nikolas Oubaid, Holger Schulz.

**Visualization:** Nikolas Oubaid.

**Writing – original draft:** Nikolas Oubaid.

**Writing – review & editing:** Nikolas Oubaid, Sukhvir Kaur, Karin Oechsle, Viola Milke, Anneke Ullrich, Aneta Schieferdecker, Kerstin Kremeike, Sophie Meesters, Christin Herrmann, Raymond Voltz, Holger Schulz.

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
