## [Decision Letter · Decision Letter 0]

26 Feb 2025

PONE-D-24-31789How do healthcare professionals on non-palliative care wards perceive quality of care in the dying phase? Personal and organizational predictors identified in a cross-sectional studyPLOS ONE?

Dear Dr. Oubaid,

Thank you for submitting your manuscript to PLOS ONE. After careful consideration, we feel that it has merit but does not fully meet PLOS ONE’s publication criteria as it currently stands. Therefore, we invite you to submit a revised version of the manuscript that addresses the points raised during the review process.

publication criteria  and not, for example, on novelty or perceived impact.

We look forward to receiving your revised manuscript.

Kind regards,

Stefan Grosek, Ph.D., M.D.,

Academic Editor

PLOS ONE

Journal Requirements:

Additional Editor Comments:

Dear Authors

It took several months to finally collect reviewers' reports. Please follow comments and reply item by ited adressed by the reviewers.

Kind regards

Reviewers' comments:

Reviewer's Responses to Questions

**Comments to the Author**

1. Is the manuscript technically sound, and do the data support the conclusions?

Reviewer #1: Yes

Reviewer #2: Partly

2. Has the statistical analysis been performed appropriately and rigorously?

Reviewer #1: Yes

Reviewer #2: No

3. Have the authors made all data underlying the findings in their manuscript fully available?

Reviewer #1: No

Reviewer #2: Yes

4. Is the manuscript presented in an intelligible fashion and written in standard English?

Reviewer #1: No

Reviewer #2: Yes

Reviewer #1: Thank you for the opportunity to review this paper. It is a well-written piece of work. However, the rationale for the choice of outcome of interest of this study is not clear.

Abstract

“This makes their perspective important to improve the quality of care in the dying phase.” – perspective on what? Please re-write this sentence.

Methods – What were the predictors?

Results – “The regression model was significant (p<0.001) and explained 30.7% of the total variance.” – This sentence is not necessary in the abstract. We don’t really know what was in the model.

Discussion and in general: There is an underlying assumption that the perception of healthcare professionals = reality, which is not the case. Why should we explore what predicts healthcare professional’s perceptions of quality of care in the dying phase? Why is it worthwhile to have interventions to improve the perceived quality of care in the dying phase, rather than interventions that can improve the quality of care measured with patient-centred outcome measures and establish quality metrics?

Introduction

Page 3, Line 88 – “their perspectives and experiences are highly important to improve quality of care in the dying phase.” – But this paper does not explore people’s perspectives on how to improve the quality of care for people at the end of life, nor their experiences.

Page 3, Line 92 – Empirical

Please use the introduction section of the manuscript to provide background literature but also to give a rationale for why this study is needed. Listing every single association with the quality of care or attitudes towards end-of-life care (which are not your outcome) is not helpful to the reader. For instance, on page 4, line 104 – “However, the perception between different professions of patient satisfaction and of the quality of care provided can differ [21, 22].” – what does ‘different professions of patient satisfaction’ mean?

Page 4, Lines 109- 110: “Palliative care-specific training showed a learning effect and an increase in perceived competence among HCPs, but it is not known whether this translates into better, sustained quality of care [26, 27].” – this paper is not measuring the quality of care.

Page 5 – Line 133 – GW unnecessary acronym.

Page 5 – Lines 133-134: “However, their perception is important, as they are also the providers of care in the dying phase.” – Why? Is there evidence of concordance between patient’s perception of quality of care and HCPs’ perceptions? Are their perceptions correlated with actual care quality that is delivered?

Page 5 – Page 135-137: What does “overall quality of care” mean? There is a massive amount of work dedicated to measuring and improving the quality of care towards the end of life. Please support these statements (which are not clear) with references if they are true.

Methods

Page 7 – Lines 174-175: If the authors do not recommend summing up the scores, why did you go ahead and do it anyway? Surely, this means adding the items up does not lead to a meaningful sum.

Data Analysis

Please state which variables were dichotomised and the reasons. Thank you for providing a detailed explanation of the statistics checks. Your final sample seems to be skewed towards people who work in ICUs. There are more general wards in hospitals than ICUs.

Results

Do you have any information on the non-respondents and the denominator for the sample?

Why did you dichotomise the profession?

Page 11, Line 249: assess not address

Page 13, fully adjusted model: It may be useful to provide the findings for the minimally adjusted model and univariate correlations as well as the fully adjusted model.

In table 6 - just checking that the labels weak and moderate associations are not based on the p value? Please make this clearer.

Discussion

The discussion section should highlight the main findings from the study, not describe the modelling steps. Please revise the first paragraph.

Page 15, line 303 – Please provide references and actually discuss what your findings may mean.

Page 15, lines 316-318: “With regard to the perceived quality of care in the dying phase, it is possible that younger HCPs compare the care provided on their ward with a more idealistic care they know from their training and education.” – Is this necessarily a bad thing? The explanation for this finding seems like a bit of a stretch.

Discussion of non-significant predictors – Another explanation can be that the previous studies did not assess the association between perceived quality of care at the end of life from the HCPs’ point of view.

Clinical implications

I am still not convinced what the implication of improving the perception of the quality of care, rather than improving the quality of care.

Reviewer #2: Introduction:

1. Recommend not starting the first line with "How".

2. In paragraphs 81–88, consider including more figures and percentages from the studies cited to provide a clearer picture of existing literature.

3. Reference is needed for line 92.

4. Line 95: Suggest rephrasing “While Spiritual care…” for better structure.

5. Personal factors section: Organize the breakdown of factors better, as the current flow is disjointed.

6. Work-related organizational factors: Needs better organization and more structured listing.

7. Lines 124–126: Expound on research regarding interprofessional teamwork and its benefits. Clarify the nuances of good interprofessional work in other studies.

Data:

1. Mention earlier in the study that it was done during the COVID epidemic, not just in limitations.

2. No power analysis: Clarify how sample size was calculated and whether it was powered to detect differences, particularly with a 35% response rate.

3. Did you perform AIC/BIC analysis to determine if the Hierarchical Regression model was the best fit? Please clarify.

4. Address checks for outliers, such as Cook’s distance.

5. Results for age (p = 0.02) show significance, but the effect size (Beta = 0.15) is small, and the confidence interval is wide, suggesting uncertainty. Did dichotomizing age with an unbalanced group affect the accuracy of this result?

6. Improve the descriptive results section by presenting medians and SDs graphically instead of as text.

7. Clarify the scale behind terms like “Moderate” and “High” in line 254–255.

8. The p-value of 0.06 between wards should be discussed more, as it is near significance. Was the sample size sufficient to detect differences?

9. Consider testing for interaction effects between nurse status and ward type (ICU vs general ward), as this could reveal if the relationship differs by setting.

Limitations:

1. Comment on the 35% response rate and its potential impact on the sample. Could there be response bias?

2. Address underrepresentation of non-nurses and the overrepresentation of ICU nurses compared to general ward nurses.

3. Suggested to revisit the study with updated data post-COVID, as findings may not be generalizable outside a pandemic context.

**Do you want your identity to be public for this peer review?** For information about this choice, including consent withdrawal, please see our Privacy Policy

Reviewer #1: No

Reviewer #2: No

---

## [Author Response · Author response to Decision Letter 1]

9 Apr 2025

I uploaded the "response to the reviewers" in the "attach files"-section.

Here is an additional copy:

Manuscript ID: PONE-D-24-31789

Manuscript’s title: How do healthcare professionals on non-palliative care wards perceive quality of care in the dying phase? Personal and organizational predictors identified in a cross-sectional study

Corresponding Author: Nikolas Oubaid

Response to Reviewers

Dear Editor, dear reviewers,

Thank you for giving my co-authors and me the opportunity to revise our manuscript titled “How do healthcare professionals on non-palliative care wards perceive quality of care in the dying phase? Personal and organizational predictors identified in a cross-sectional study”. My co-authors and I appreciate the time and effort that you have dedicated to providing and communicating valuable feedback on our manuscript and suggesting ideas for improvement on a detailed level.

We incorporated changes to reflect most of the suggestions provided by the reviewers and we highlighted the changes within the manuscript (red text for additions, crossed-out gray text for deletions).

We responded to every aspect of the reviewer’s feedback point-by-point:

Reviewer #1:

Overall response: Thank you for your detailed feedback, especially on our main outcome and the rationale of our work, helping us to improve the overall quality and readability of our manuscript. We substantially revised the “introduction” and “discussion” chapters regarding your and Reviewer #2’s comments. We also added a comprehensive “Analysis Appendix”. Thank you for pointing out some typographical errors – we revised the entire manuscript for readability.

Point-by-point response:

Abstract

“This makes their perspective important to improve the quality of care in the dying phase.” – perspective on what? Please re-write this sentence.

Thank you for this hint, we rephrased that sentence and clarified, that we are investigating HCPs’ perceptions of the quality of care in the dying phase on their ward (page 2, line no. 55-56).

Methods – What were the predictors?

We added the predictor variables to the abstract (page 2, line no. 63-66). Thank you for this hint!

Results – “The regression model was significant (p<0.001) and explained 30.7% of the total variance.” – This sentence is not necessary in the abstract. We don’t really know what was in the model.

Thank you for this hint! As we added the predictors to the abstract like you suggested, therefore we let this sentence remain in the abstract.

Discussion and in general: There is an underlying assumption that the perception of healthcare professionals = reality, which is not the case. Why should we explore what predicts healthcare professional’s perceptions of quality of care in the dying phase? Why is it worthwhile to have interventions to improve the perceived quality of care in the dying phase, rather than interventions that can improve the quality of care measured with patient-centred outcome measures and establish quality metrics?

Thank you for sharing this very reasonable concern! We agree and there are studies that imply that the perceived quality of care of HCPs does not always equal the quality of other indicators (e.g. quality reported by patients or informal caregivers). However, the perception of HCPs is a significant parameter to improve quality of care, especially in the dying phase (many patients in the dying phase are non-communicative and patient reported outcomes are difficult to measure and collect). HCPs also often experience moral distress when caring for dying patients or experience higher levels of distress in general, when they perceive the quality of a patients’ death as low. HCPs’ perception is therefore important to create a healthy work environment, keep HCPs engaged and consequently to improve the quality of care in the dying phase.

However, as we see your valuable point we revised the entire “introduction” chapter and added a stringent paragraph regarding the importance of HCPs’ perception of the quality of care – especially in the dying phase (page 5-6, line no. 146-178).

Introduction

Page 3, Line 88 – “their perspectives and experiences are highly important to improve quality of care in the dying phase.” – But this paper does not explore people’s perspectives on how to improve the quality of care for people at the end of life, nor their experiences.

Thank you for pointing this out! We agree that our original reasoning is not specific enough. We rephrased that sentence (and the rest of the “introduction” chapter) to focus on why HCPs’ perception of the quality of care in the dying phase is an important parameter for the healthcare system and to improve quality of care (pages 5-6, line no. 146-178). We also clarified that our outcome is not perspective nor experience, but perception of quality of care in the dying phase.

Page 3, Line 92 – Empirical

Thank you for this hint – we corrected the word (page 4, line 109). We also spell checked the rest of the document.

Please use the introduction section of the manuscript to provide background literature but also to give a rationale for why this study is needed. Listing every single association with the quality of care or attitudes towards end-of-life care (which are not your outcome) is not helpful to the reader. For instance, on page 4, line 104 – “However, the perception between different professions of patient satisfaction and of the quality of care provided can differ [21, 22].” – what does ‘different professions of patient satisfaction’ mean?

Thank you for this comment - in context of your overarching comment regarding the outcome and the question of whether the perception of quality of care is relevant and in combination with comments from Reviewer #2, we have generally revised the “introduction” chapter (also see your final comment “clinical implications”) and enhanced the part about our rationale (page 5-6, line no. 146-178). However, we have retained some empirical studies and reviews of the literature on the predictors of quality of care, as we assessed if these discussed predictors of quality of care (in the dying phase), mostly reported by patients and informal caregivers, are also predictors of HCPs’ perception. As we see your point, we revised the original paragraph about the predictor variables as well (pages 4-5, line no. 109-138).

Regarding the “different professions”-comment: We revised that sentence (page 4, line no. 112-113). Thank you for pointing this out!

Page 4, Lines 109- 110: “Palliative care-specific training showed a learning effect and an increase in perceived competence among HCPs, but it is not known whether this translates into better, sustained quality of care [26, 27].” – this paper is not measuring the quality of care.

Thank you for this hint! As we revised the “Introduction” chapter, we also revised this paragraph/sentence accordingly (page 5, line no. 136-138).

Page 5 – Line 133 – GW unnecessary acronym.

We deleted GW as an acronym to increase readability of the manuscript. However, we left it in figures and tables to save space and increase an easy overview of the data.

Page 5 – Lines 133-134: “However, their perception is important, as they are also the providers of care in the dying phase.” – Why? Is there evidence of concordance between patient’s perception of quality of care and HCPs’ perceptions? Are their perceptions correlated with actual care quality that is delivered?

Thank you for this important hint and the question! We revised the entire “Introduction” chapter and pointed out, why HCPs’ perception of quality of care in the dying phase is an important parameter and how the perception of quality of care is embedded in clinical practice (page 5-6, line no. 146-178).

We also added literature to support the statement that HCPs’ perception of quality of care in the dying phase is a common method to measure quality of care in the dying phase (e.g. a systematic review by Kupeli et al. (2019) mentions the challenges of measuring quality of care in the dying phase and lists tools that are also based on HCPs’ perceptions).

Page 5 – Page 135-137: What does “overall quality of care” mean? There is a massive amount of work dedicated to measuring and improving the quality of care towards the end of life. Please support these statements (which are not clear) with references if they are true.

Thank you for this reasonable concern! We used this self-constructed item (overall quality of care) intentionally, because in palliative care, certain single aspects of care (psychological care, intensity of pain or other somatic symptoms, …) are often being used as outcomes (also in the literature we cited) and are part of long and complex item sets. We used this single item as a proxy-variable. We hoped that using a short and single item compared to a longer item set would also increase the motivation of HCPs to finish the survey (out of the 35% response rate, we managed to receive a 79% completion rate).

We also added a statement to the “data analysis” chapter on why we intentionally used the self-constructed item (pages 9-10, line no. 243-246).

As this study needs to be revised on a larger, independent sample we also suggested in the “future research” chapter that this item should be used along established item sets for further validation (page 21, line no. 452-460).

Methods

Page 7 – Lines 174-175: If the authors do not recommend summing up the scores, why did you go ahead and do it anyway? Surely, this means adding the items up does not lead to a meaningful sum.

Thank you for this methodically important question! We rephrased that sentence (page 8, line no. 209-210). In our opinion, the single items of this instrument provide a good representation of the subjectively perceived burden factors that can occur when caring for dying patients and supporting their informal caregivers as the items represent a checklist of different aspects of burden factors. Additionally, its’ Cronbach’s Alpha value is above 0.75 in our data as well. Furthermore, these items have already been used on palliative care wards in a larger national study.

However, the point you are making is very important, and this measurement instrument would need to be validated further. We have therefore added a corresponding sentence to the “discussion of non-significant predictors” (page 18, line no. 395-397) and the “strengths and limitations” (page 20, line no. 426-427) chapters.

Data Analysis

Please state which variables were dichotomised and the reasons. Thank you for providing a detailed explanation of the statistics checks. Your final sample seems to be skewed towards people who work in ICUs. There are more general wards in hospitals than ICUs.

Thank you for this suggestion and the hint! We clarified which variables were dichotomized (page 12, line no. 274). We also gave specific reasons (page 12, line no. 276-278).

We expanded our limitations regarding the imbalance of ICU and general ward HCPs in our sample (page 19, line no. 422-424). We also added to the “future research” chapter that this study should not only be repeated outside of pandemic context, but also with a more balanced sample (pages 20-21, line no. 454-457).

We also included an “Analysis Appendix” for further statistical background information.

Results

Do you have any information on the non-respondents and the denominator for the sample?

Thank you for this methodological important question! We have provided available and relevant information regarding non-responders and the samples’ denominator in the Appendix (STROBE-Guideline and CHERRIES-Guideline).

Why did you dichotomise the profession?

Thank you for this appropriate question. We dichotomized profession because studies suggest that nurses tend to perceive the quality of care different than other professions. Nurses are also more than 50% of our sample, so we wanted to proceed with a balanced comparison. We added a corresponding sentence (pages 12, line no. 276-278).

Page 11, Line 249: assess not address

Thank you for this hint – we corrected the word (page 13, line no. 289).

Page 13, fully adjusted model: It may be useful to provide the findings for the minimally adjusted model and univariate correlations as well as the fully adjusted model.

Thank you for this great suggestion! We added a comprehensive “Analysis Appendix” (first reference in the manuscript: page 13, line no. 296-297) where we displayed bivariate correlations of the predictor variables and the outcome variable, as well as the minimal model (step one, model without organizational predictors). Readers can therefore now better understand correlational relationships between the predictor variables as well as the two regression models.

We also added the results of the AIC analysis (page 14, line no. 310-311) to give the reader a better understanding of the model selection process.

In table 6 - just checking that the labels weak and moderate associations are not based on the p value? Please make this clearer.

Thank you for the good idea to clarify that by “weak”, “moderate” and “strong” we refer to the standardized beta-values. We made this clear by adding the beta-values (pages 15-16, table 6).

Discussion

The discussion section should highlight the main findings from the study, not describe the modelling steps. Please revise the first paragraph.

Thank you for this correct hint! We revised the entire beginning of our “discussion” chapter (page 16, line no. 342-348).

Page 15, line 303 – Please provide references and actually discuss what your findings may mean.

Thank you for this comment, we added a paragraph regarding possible implications of our results (page 17, line no. 355-360).

Page 15, lines 316-318: “With regard to the perceived quality of care in the dying phase, it is possible that younger HCPs compare the care provided on their ward with a more idealistic care they know from their training and education.” – Is this necessarily a bad thing? The explanation for this finding seems like a bit of a stretch.

Thank you for noticing this! We revised the wording in a more neutral way (page 17, line no. 371-372). In the context of Reviews #2s’ comment regarding this predictor, we also added a sentence regarding the impact of this predictor (page 18, line no. 375-377).

Discussion of non-significant predictors – Another explanation can be that the previous studies did not assess the association between perceived quality of care at the end of life from the HCPs’ point of view.

Thank you for this valuable hint! We added a corresponding sentence (page 19, line no. 402-405) and addressed this aspect.

Clinical implications

I am still not convinced what the implication of improving the perception of the quality of care, rather than improving the quality of care.

We rephrased the manuscript to emphasise why it is important to improve HCPs’ perception of quality of care in the dying phase (see your previous comments). We hope that our revision of the manuscript resolved your concern.

Reviewer #2:

Overall response: Thank you for your feedback, especially on the readability of our manuscript, as well as on methods and data analysis. Thank you for suggesting the reporting of additional tests to improve the methodological quality of our manuscript. Regarding Reviewer #1 and your comments we also added a comprehensive “Analysis Appendix”, providing additional information.

Point-by-point response:

Introduction

1. Recommend not starting the 1st line of the paragraph with “How”.

a. Thank you for this suggestion, we rephrased the beginning of our manuscript (page 3, line no. 87-88).

2. Paragraphs 81 - 88. Would there be a possibility to add more of what the studies had shown to give the reader a more clear picture of the known literature on this topic by including percentages/figures.

a. This is a very reasonable request! In the context of your other comments regarding the introduction and Reviewer #1’s comments, we decided to revise the entire “introduction” chapter. We emphasised why HCPs’ perception of quality of care is an important clinical outcome and that we wanted to check whether the discussed predictors of quality of (end-of-life) care are also predictors of HCPs’ perceived quality of care in the dying ph

---

## [Decision Letter · Decision Letter 1]

18 Aug 2025

PONE-D-24-31789R1How do healthcare professionals on non-palliative care wards perceive quality of care in the dying phase? Personal and organizational predictors identified in a cross-sectional studyPLOS ONE?

Dear Dr. Oubaid,

Thank you for submitting your manuscript to PLOS ONE. After careful consideration, we feel that it has merit but does not fully meet PLOS ONE’s publication criteria as it currently stands. Therefore, we invite you to submit a revised version of the manuscript that addresses the points raised during the review process.

The authors need to correct or accept some minor issues.

Please read the reviewers' comments and correct them where you see fit.

We look forward to receiving your revised manuscript.

Kind regards,

Stefan Grosek, Ph.D., M.D.,

Academic Editor

PLOS ONE

Journal Requirements:

Additional Editor Comments:

Dear Authors

Finally, I received comments from the reviewers. Mos of the raised issues were solved, some minor issues to be amended.

Kind regards

Reviewers' comments:

Reviewer's Responses to Questions

**Comments to the Author**

Reviewer #2: All comments have been addressed

Reviewer #3: All comments have been addressed

Reviewer #4: All comments have been addressed

2. Is the manuscript technically sound, and do the data support the conclusions?

Reviewer #2: Yes

Reviewer #3: Yes

Reviewer #4: Yes

3. Has the statistical analysis been performed appropriately and rigorously?

Reviewer #2: Yes

Reviewer #3: Yes

Reviewer #4: Yes

4. Have the authors made all data underlying the findings in their manuscript fully available?

Reviewer #2: Yes

Reviewer #3: Yes

Reviewer #4: Yes

5. Is the manuscript presented in an intelligible fashion and written in standard English?

Reviewer #2: No

Reviewer #3: Yes

Reviewer #4: Yes

Reviewer #2: Overall much improvement is noted in this final revision with significant impact regarding the methodology rigor which represents the backbone of this study. Some small grammar errors at noted regarding "Working on ICU" in multiple sections and also not including the power analysis which needs to be commented on. With small minor revisions left, I am impressed with the work placed to pull this paper to be much better

Reviewer #3: (No Response)

Reviewer #4: The authors present in the manuscript an interesting and less known area of self-assessment of healthcare professionals in palliative care of dying patients. They conducted a survey of HCP on ten non-palliative care hospital wards and of two university medical centers and tested HCP perception of quality of care in the dying phase as well as its predictors. They found that perceived quality of interprofessional patient-centered teamwork was the most clinically relevant as well as modifiable predictor of perceived quality of care in the dying phase could be beneficial for clinical outcomes.

The article has, thanks to the corrections already made and suggested by the previous reviewers, gained a great deal in quality in all the chapters. The statistics of the article are excellently set, sophisticated.

Now I propose minor corrections for better understanding in the revised manuscript:

- Line 131: the term ”predictor of outcome” should be changed to more appropriate expression (like: predictors of quality of care at the end of life”)

- Line 138: the “clinical outcome” should be changed

- Line 140: “quality of patients death” should be changed to more appropriate expression

- Line 161-163: the sentence ”Monitoring HCP perception………………….HCP well-being” should be changed as it is not understandable.

- Line 258: Outcome variable should be specified (HCP perceived quality of care in the dying?)

- Line 359: I advise to use more appropriate expression for “outcome measures”

- Line 360: I advise to use more appropriate expression for “outcome

- Line 364: Expression “outcome variables” should be change to “variables”

- Line 404: I advise to use more appropriate expression for “outcome” measures

- Line 405: I advise to use more appropriate expression for caregivers reported “outcome”

- Line 409: I advise to use more appropriate expression for “outcome”

**Do you want your identity to be public for this peer review?** For information about this choice, including consent withdrawal, please see our Privacy Policy

Reviewer #2: No

Reviewer #3: No

Reviewer #4: **Yes: ** Andreja Sinkovic

---

## [Author Response · Author response to Decision Letter 2]

16 Sep 2025

My response to the editor and reviewers is also uploaded in an additional file.

Dear Editor, dear Reviewers,

Thank you for giving my co-authors and me the opportunity to revise our manuscript titled “How do healthcare professionals on non-palliative care wards perceive quality of care in the dying phase? Personal and organizational predictors identified in a cross-sectional study”. My co-authors and I appreciate the time and effort that you have dedicated to providing and communicating valuable feedback on our revised manuscript and suggesting ideas for improvement on a detailed level.

We incorporated changes to reflect most of the suggestions provided by the reviewers, and we highlighted the changes within the manuscript (red text for additions, crossed-out gray text for deletions).

Overall: We have cited a paper that provides a more detailed analysis of the burden factors (page 6, line no. 163-164).

Reviewer #2:

Overall comment: Overall much improvement is noted in this final revision with significant impact regarding the methodology rigor which represents the backbone of this study. Some small grammar errors at noted regarding "Working on ICU" in multiple sections and also not including the power analysis which needs to be commented on.

With small minor revisions left, I am impressed with the work placed to pull this paper to be much better.

Overall response: Thank you very much for taking the time to review our original submission and our revised version. We will address your specific comments below.

Point-by-point response:

Introduction

1. Written with a large amount of expansion on the background evidence. Writing style remains cumbersome with a vast amount of information which can be concisely written

2. Recommend Highlighting the points which would mirror what is being studied in the paper which will allow smooth flow and exposing which factors the reader should be paying attention to while reading

3. Recommend condensing the introduction as it appears almost in length as the discussion.

a. Thank you for your comments and suggestions! The “Introduction” was most affected by the first round of revisions, which is why it is indeed cumbersome in some places.

Based on your feedback, we have now shortened and condensed the introduction (pages 3-5. Line no. 85-130 in the clean version), while also still considering the original comments from Reviewer #1 (e.g. the necessity of HCPs’ perception). The introduction has been shortened to two pages in total. We also reduced the amount of literature we cite to increase readability and focus on the most relevant aspects.

4. Recommend defining thanatophobia as well

a. Thank you for this good idea – we defined thanatophobia as fear of death and dying (page 4, line no. 113-114).

Statistical Analysis

1. Please comment on the power analysis done for the study

a. Thank you for this valuable aspect! We did not initially perform a power analysis for this specific statistical model (Kremeike et al. 2022, Study Protocol), which was a linear regression model with quality of care as the independent variable and 12 predictor variables. However, we specified that we expected a response rate for the staff survey of 50% (Kremeike et al. 2022) (page 7, line no. 173-174). Since we managed to achieve a response rate of 35% (reasons for non-participation are listed in the STROBE-Guideline), we mentioned that we did not reach our expected sample size in our “data collection”-chapter (page 7, line no. 185-186) and in our “strengths and limitations”-chapter (page 17, line no. 359-360).

b. We also conducted a post-hoc power analysis for the predictor “type of ward” and the outcome “quality of care”: The analysis showed that with the given sample size (ICUs n=126, GWs n=75) and a power of ≥80% and a two-sided alpha of 5% we were able to detect a standardized effect (Cohen's d) of d≥0.41 with unpaired t-test statistics. In our regression model, the transformed effect size (unstandardized B coefficient / standard error of the estimate) for “type of ward” was d=0.31, which unfortunately fell slightly below d=0.41 and thus the “type of ward” factor is not significant. We mentioned this analysis in the “data collection”-chapter (page 7, line no. 186) and added the calculation to the STROBE-Guideline appendix (S2).

Results

1. Line 240. Recommend rephrasing “Worked on ICUs”.

a. Thank you for this stylistic hint. We rephrased that in several chapters (e.g. page 17, line no. 358 and 359; page 4, line no. 108; page 9, line no. 222 and 223).

Limitations

- Low Response rate should be mentioned as a limitation as well

a. Thank you for this suggestion – we incorporated a corresponding sentence in “strengths and limitations” (page 17, line 359-360).

Additional Comment

- Improved from prior original version with much more detailed and rigorous statistical analysis

- Clear explanations of the limitations noted and context of the study which is helpful

Reviewer #3:

Overall response: Thank you for reading and assessing our revised manuscript!

Point-by-point response:

There were no specific issues raised by Reviewer #3.

Reviewer #4:

Overall comment: The article has, thanks to the corrections already made and suggested by the previous reviewers, gained a great deal in quality in all the chapters. The statistics of the article are excellently set, sophisticated.

Overall response: Thank you very much for agreeing to be an additional reviewer and for reading and evaluating our revised version. Thank you also for appreciating the work that went into the statistical analysis.

Thank you for your comments on the terminology! Due to the suggestion of Reviewer #2, we shortened and rewrote the introduction. Consequently, we deleted some of the sentences you suggested rephrasing. However, we rephrased the remaining aspects you mentioned.

We will address your specific concerns below.

Point-by-point response:

• Line 131: the term ”predictor of outcome” should be changed to more appropriate expression (like: predictors of quality of care at the end of life”)

- Thank you for this suggestion – we altered the sentence.

• Line 138: the “clinical outcome” should be changed

- We changed “clinical outcome” to HCPs’ perceived quality of care.

• Line 140: “quality of patients death” should be changed to more appropriate expression

- This sentence got deleted in the context of the comments made by Reviewer #2.

• Line 161-163: the sentence ”Monitoring HCP perception………………….HCP well-being”

- This sentence got deleted in the context of the comments made by Reviewer #2.

• Line 258: Outcome variable should be specified (HCP perceived quality of care in the dying?)

- Thank you for the suggestion! We changed “outcome variable” to HCPs’ perceived quality of care.

• Line 359: I advise to use more appropriate expression for “outcome measures”

- Thank you for comment. We changed “outcome” to “HCPs’ perceived quality of care”.

-

• Line 360: I advise to use more appropriate expression for “outcome”

- Thank you for comment. We changed “outcome” to “HCPs’ perceived quality of care”.

• Line 364: Expression “outcome variables” should be change to “variables”

- Thank you for your specific suggestion. We altered the terminology accordingly.

-

• Line 404: I advise to use more appropriate expression for “outcome” measures

- Thank you for your comment – we changed “outcome measure” to “dependent variable”.

-

• Line 405: I advise to use more appropriate expression for caregivers reported “outcome”

- Thank you for this hint! We now use “Informal caregivers’ perception”.

• Line 409: I advise to use more appropriate expression for “outcome”

- Thank you for comment. We changed “outcome” to “HCPs’ perceived quality of care”.

---

## [Decision Letter · Decision Letter 2]

1 Oct 2025

How do healthcare professionals on non-palliative care wards perceive quality of care in the dying phase? Personal and organizational predictors identified in a cross-sectional study

PONE-D-24-31789R2

Dear Dr. Oubaid,

We’re pleased to inform you that your manuscript has been judged scientifically suitable for publication and will be formally accepted for publication once it meets all outstanding technical requirements.

Kind regards,

Stefan Grosek, Ph.D., M.D.,

Academic Editor

PLOS ONE

Additional Editor Comments (optional):

ar Authors

All comments have been addressed. Thank you for you submission. I'll recommend for acceptance and publication.

Kind regards

Reviewers' comments:

Reviewer's Responses to Questions

**Comments to the Author**

Reviewer #2: All comments have been addressed

Reviewer #4: All comments have been addressed

2. Is the manuscript technically sound, and do the data support the conclusions?

Reviewer #2: Yes

Reviewer #4: Yes

3. Has the statistical analysis been performed appropriately and rigorously?

Reviewer #2: Yes

Reviewer #4: Yes

4. Have the authors made all data underlying the findings in their manuscript fully available?

Reviewer #2: Yes

Reviewer #4: Yes

5. Is the manuscript presented in an intelligible fashion and written in standard English?

Reviewer #2: Yes

Reviewer #4: Yes

Reviewer #2: This paper has improved regarding the grammar and sentence structure as well as clarity. The statistical analysis is much more solid and clear. Happy to see the development and forward progress on the background of the hard work done by the research team

Reviewer #4: The authors responded to all the comments of the reviewers and changed the manuscript accordingly. In this way the manuscript can be published

**Do you want your identity to be public for this peer review?** For information about this choice, including consent withdrawal, please see our Privacy Policy

Reviewer #2: No

Reviewer #4: **Yes: ** Andreja Sinkovic

---

## [Editor Report · Acceptance letter]

PONE-D-24-31789R2

PLOS ONE

Dear Dr. Oubaid,

I'm pleased to inform you that your manuscript has been deemed suitable for publication in PLOS ONE. Congratulations! Your manuscript is now being handed over to our production team.

Kind regards,

on behalf of

Professor Stefan Grosek

Academic Editor

PLOS ONE